# Unbiased measurements of reconstruction fidelity of sparsely sampled magnetic resonance spectra

Qinglin Wu[1], Brian E. Coggins[1] & Pei Zhou[1]

The application of sparse-sampling techniques to NMR data acquisition would benefit from reliable quality measurements for reconstructed spectra. We introduce a pair of noise-normalized measurements, $R_{\text{work}}^{\text{noise}}$ and $R_{\text{free}}^{\text{noise}}$, for differentiating inadequate modelling from overfitting. While $R_{\text{work}}^{\text{noise}}$ and $R_{\text{free}}^{\text{noise}}$ can be used jointly for methods that do not enforce exact agreement between the back-calculated time domain and the original sparse data, the cross-validation measure $R_{\text{free}}^{\text{noise}}$ is applicable to all reconstruction algorithms. We show that the fidelity of reconstruction is sensitive to changes in $R_{\text{free}}^{\text{noise}}$ and that model overfitting results in elevated $R_{\text{free}}^{\text{noise}}$ and reduced spectral quality.

[1] Department of Biochemistry, Duke University Medical Center, Durham, North Carolina 27710, USA. Correspondence and requests for materials should be addressed to P.Z. (email: peizhou@biochem.duke.edu).

Recent developments in sparse-sampling and iterative reconstruction techniques have made it possible to acquire magnetic resonance spectra using a fraction of the measurement time required by the conventional Nyquist-sampling method, without sacrificing spectral resolution or quality[1–5]. Despite these remarkable advancements, there are not, as yet, unbiased, quantitative measurements of the fidelity of reconstructed spectra. Instead, the quality of spectral reconstruction is frequently assessed by direct comparison with artifact-free spectra generated from fully sampled datasets, by examination of algorithm-specific parameters, or by estimating the reduction of aliasing artifacts in the reconstructed spectra. Each of these measurements has its own limitations: comparison of reconstructed spectra with artifact-free spectra from fully sampled data is not feasible in real applications; spectra reconstructed with different methods cannot be compared using algorithm-specific parameters; and finally, excessive 'artifact' reduction during reconstruction may not correlate with the improvement of spectral fidelity.

Here, we introduce two algorithm-independent measurements for evaluating the quality of nuclear magnetic resonance (NMR) spectra reconstructed from sparsely sampled datasets, demonstrate their utility in differentiating inadequate modelling from overfitting, and discuss the implication of such quality measurements for the fidelity of NMR spectral reconstruction.

## Results

**Quality measurements for reconstructed NMR spectra.** NMR time domain data and frequency domain spectra are connected through the Fourier transform. Conceptually, the quality of a reconstructed spectrum can be measured by computing the inverse Fourier transform of the spectrum and comparing the resulting time domain data with the raw measurements at the sampled positions. However, this alone is inadequate, as the following example illustrates. The spectrum generated from the Fourier transform of the sparsely sampled time domain data would fulfil such a criterion, yet it is not a high-quality reconstruction due to the presence of strong aliasing artifacts, which arise from the lack of modelled signals at the unmeasured positions in the time domain. A more useful quality measure would go further, flagging this as a poor reconstruction.

Similar issues were encountered previously in X-ray crystallography, where the diffraction pattern, which is related to the electron density by a Fourier transform, contains intensity information but not phase information; the phases must be reconstructed iteratively in reciprocal space as the model of the molecular structure is assembled in real space. Initially, it was proposed that the inverse Fourier transform of the modelled electron density could be compared with the diffraction data, and their correlation—in the form of the 'R-factor,' $R = \frac{\sum ||F_{\text{obs}}| - |F_{\text{calc}}||}{\sum |F_{\text{obs}}|}$, where $F_{\text{obs}}$ and $F_{\text{calc}}$ represent observed and back-calculated structure factors—would reflect the quality of the modelled electron density map[6]. If the iterative process of assembling the model is successful, its inverse Fourier transform will come increasingly close to agreeing with the observed data, and $R$ will become progressively smaller.

While $R$ is helpful, it was soon realized that $R$ alone is inadequate, as the incorporation of experimental noise into the model will drive down the R-factor while reducing, rather than improving, the fidelity of the structural model[7]. To address this, it was suggested that a small percentage of the measurements (5–10%) be set aside and excluded from the reconstruction process, and that the electron density be built from the remaining measurements, known as the working set. As the reconstruction proceeds, the consistencies of the calculated structure factors with the working dataset ($R_{\text{work}}$) and the excluded dataset ($R_{\text{free}}$) are used together to evaluate the quality of the electron density map. While incorporation of noise into the model improves the agreement with the working set (reducing $R_{\text{work}}$), it results in worse fitting to the excluded set (increasing $R_{\text{free}}$), thus allowing over-refinement to be detected[7].

Despite this conceptual similarity between the data processing of sparsely sampled NMR and crystallography, several issues must be addressed before these quality measurements can be applied to NMR. One important consideration is that some reconstruction algorithms explicitly or implicitly require exact agreement between the back-calculated time domain and the experimental measurements, resulting in an $R$ (or $R_{\text{work}}$) value of zero. While this would seem to negate the value of this approach, we show below that if the raw time domain measurements are divided into a working dataset used for reconstruction and a free dataset reserved for cross validation, the $R_{\text{free}}$ can be used alone as a meaningful measure of reconstruction quality.

The application of R-factor measurements to NMR is additionally confounded by a second distinction between NMR and crystallography. In crystallography, the occupancy of the unit cell by protein does not vary significantly, regardless of the protein under study. Therefore, the R-factors always fall into the same numeric range for crystals of a similar quality. In NMR, however, each position on the directly observed dimension of a spectrum constitutes an independent reconstruction problem, and the set of independent 2-D planes in a 3-D spectrum (or 3-D cubes in a 4-D spectrum) contain vastly different numbers of signals, from pure noise to large arrays of signals with different intensities and lineshapes. For a noise plane, the ideal reconstruction would contain no signal, the back-calculated time domain data would be zero, and both $R_{\text{work}}$ and $R_{\text{free}}$ would be 100%; whereas for a plane containing signals, the back-calculated time domain from the ideal reconstruction would be very close to the raw measurements, and the corresponding R-factors would be vanishingly small. In order to obtain a consistent readout that is independent of the number of signals involved in the reconstruction, we introduce noise-normalized quality measurements $R_{\text{work}}^{\text{noise}}$ and $R_{\text{free}}^{\text{noise}}$, which are defined as:

$$R^{\text{noise}} = \frac{\sum \left| \overrightarrow{v_{\text{obs}}} - \overrightarrow{v_{\text{calc}}} \right|}{\sum \left| \overrightarrow{v_{\text{noise}}} \right|}. \tag{1}$$

In equation (1), $\left| \overrightarrow{v_{\text{noise}}} \right|$ is the vector length of the hypercomplex measurements in a reference noise plane or cube, while $\left| \overrightarrow{v_{\text{obs}}} - \overrightarrow{v_{\text{calc}}} \right|$ is the vector difference of the observed and back-calculated time-domain signals from the model spectrum.

**Application of quality measurements to reconstructed spectra.** In order to illustrate the utility of our proposed measurements, we first used the CLEAN algorithm[8] to reconstruct the 3-D HNCO spectra of six proteins at a sampling density of 5% (Fig. 1; Supplementary Figs 1–5). Ten per cent of the measurements were excluded from spectral reconstruction and marked for $R_{\text{free}}^{\text{noise}}$ calculation. CLEAN builds a model of the frequency domain spectrum through the iterative identification of signal components, and thus $R_{\text{work}}^{\text{noise}}$ and $R_{\text{free}}^{\text{noise}}$ are naturally suited to monitoring the progress of the reconstruction. All results presented here show the model only, without the inclusion of any residuals.

Two N–CO planes were selected from the HNCO spectrum of GB1 for illustration, one containing strong signals (Fig. 1a,b) and one containing weak signals (due to leakage from the neighbouring plane, Fig. 1c,d). For both cases, as the threshold for inclusion of signals in the model decreased, individual signals were picked

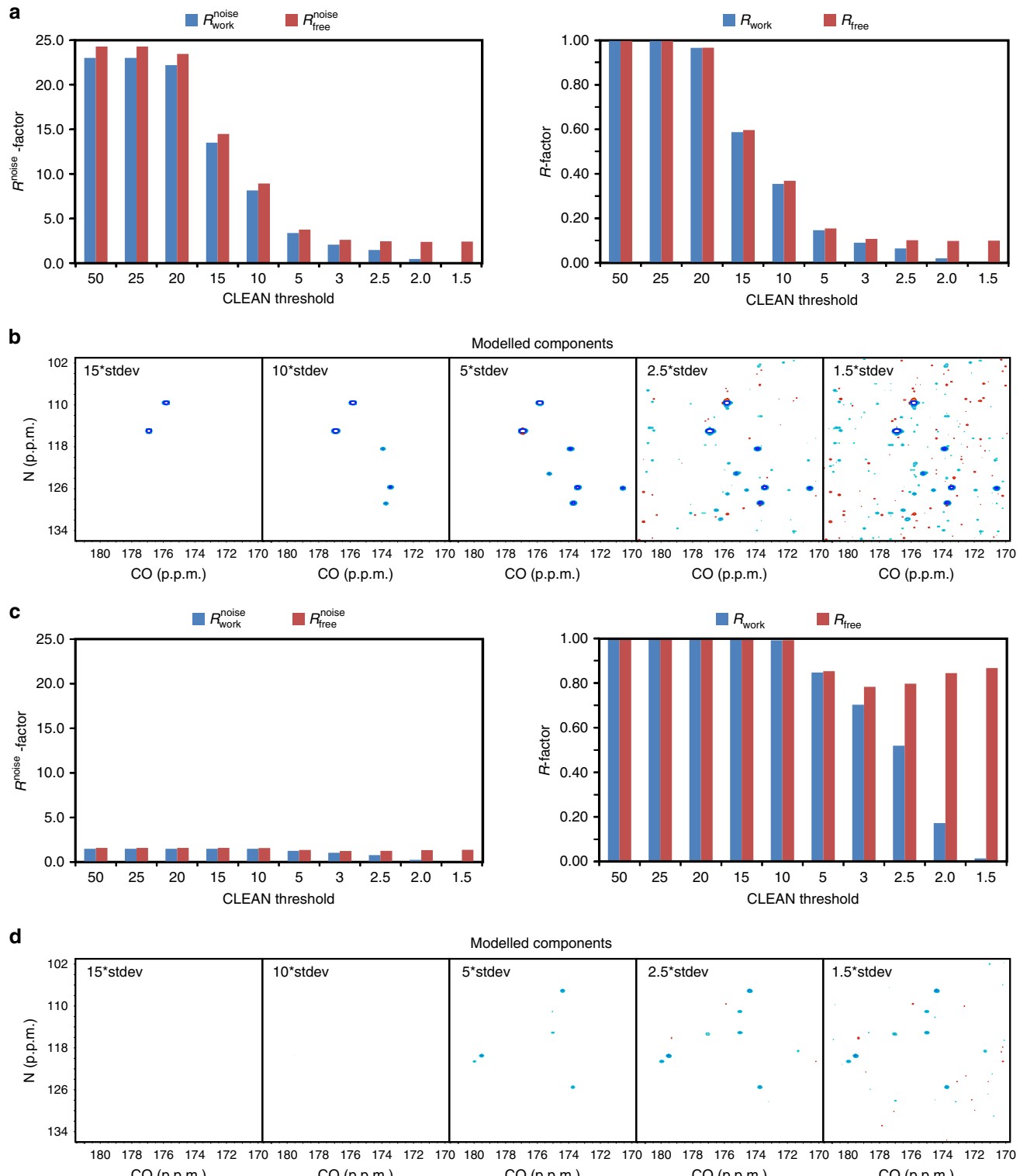

**Figure 1 | Quality measurements of selected N–CO planes from the reconstructed 3-D HNCO spectrum of GB1 by CLEAN.** (**a**) Progression of the quality factors $R^{noise}$ and $R$ during the CLEAN reconstruction of an N–CO plane containing strong signals at HN of 7.90 p.p.m. (**b**) Modelled signal components by CLEAN with different stopping thresholds for the N–CO plane at HN of 7.90 p.p.m. (**c**) Progression of quality factors $R^{noise}$ and $R$ during the CLEAN reconstruction of an N–CO plane containing weak signals at HN of 7.50 p.p.m. (**d**) Modelled signal components by CLEAN with different stopping thresholds for the N–CO plane at HN of 7.50 p.p.m.

up and incorporated (Fig. 1b,d), leading to an initial decrease of $R^{noise}_{work}$ and $R^{noise}_{free}$ values. When nearly all of the signals were modelled, the difference between the raw data and the back-calculated data was on par with the noise, with the $R^{noise}_{work}$ and

$R^{noise}_{free}$ values approaching unity. When the stopping threshold was set well below the fluctuation of aliasing artifacts, the CLEAN algorithm started to identify noise spikes and model them as signals (Fig. 1b,d). Such excessive modelling continued to

diminish $R_{\text{work}}^{\text{noise}}$, whereas $R_{\text{free}}^{\text{noise}}$ went through a minimum and then increased slightly before reaching a plateau. Such an effect was particularly noticeable for the reconstructed planes with a weaker signal-to-noise ratio (Fig. 1c). It can also be appreciated that despite similar reconstruction qualities, conventional $R$-factors normalized against signals would appear smaller for the N–CO plane containing strong signals, whereas they would seem larger for a plane containing weak signals (compare the right panels of Fig. 1a with Fig. 1c). The introduction of $R^{\text{noise}}$ factors overcomes this limitation and offers consistent measurement of the reconstruction quality: an ideal reconstruction would have $R_{\text{work}}^{\text{noise}}$ and $R_{\text{free}}^{\text{noise}}$ approaching unity regardless of the signal content. Improvement of the reconstruction is reflected by simultaneous reduction of $R_{\text{work}}^{\text{noise}}$ and $R_{\text{free}}^{\text{noise}}$, whereas overfitting results in divergent values.

We next examined whether our proposed quality measurements $R_{\text{work}}^{\text{noise}}$ and $R_{\text{free}}^{\text{noise}}$ can be applied to other reconstruction algorithms beyond CLEAN. In order to demonstrate the general applicability of these quality measurements, we implemented three popular reconstruction algorithms: convex $l_1$-norm minimization[9], maximum entropy reconstruction[10,11], and iterative soft thresholding (IST)[12].

Convex $l_1$-norm minimization, commonly used in compressed sensing, optimizes the frequency domain data to generate the spectrum with the smallest possible $l_1$-norm while having the inverse Fourier transform be consistent with the experimental time domain measurements. Optimizing against both measures is possible through a constrained minimization in which a Lagrangian multiplier $\lambda$ is introduced to balance the two requirements:

$$C = L_1(S) + \lambda \times \text{RMSD}(s - m). \tag{2}$$

In equation (2), $C$ is the composite score, $S$ is the modelled frequency domain spectrum, $s$ is the modelled time domain signals, $m$ is the experimental measurements, and RMSD is the root mean square deviation.

In order to examine the behaviour of our quality measurements for reconstruction by $l_1$ minimization, we generated a simulated sparsely sampled 1-D time domain dataset, which was Fourier transformed to yield an initial spectrum containing aliasing artifacts (Fig. 2a). A reference spectrum was also generated that contained the fully sampled signals and noise (Fig. 2b). Before reconstruction, the time domain measurements were separated into two parts, the working dataset used for reconstruction and the free dataset reserved for cross validation.

We examined the $R_{\text{work}}^{\text{noise}}$ and $R_{\text{free}}^{\text{noise}}$ values of each reconstruction as a function of the Lagrangian multiplier ($\lambda$), which alters the amount of weight placed on regularization versus agreement with the experimental data. At each value of $\lambda$, the reconstruction was obtained at the minimum of the composite score. With $\lambda$ set to zero, the scoring function consists solely of the $l_1$-norm, and minimization of this norm drives the frequency domain spectrum to the baseline, yielding a final score of zero at the end of the reconstruction process (Fig. 2c). As $\lambda$ is increased, putting more weight on the consistency between the modelled time domain signals $s$ and the experimental measurements $m$, more signals are retained in the reconstruction and the final composite score increases (Fig. 2c); at the same time, the values for both $R_{\text{work}}^{\text{noise}}$ and $R_{\text{free}}^{\text{noise}}$ begin to decrease (Fig. 2d). While the $R_{\text{work}}^{\text{noise}}$ at the end of reconstruction continues to decrease with ever-increasing values of $\lambda$, eventually reaching zero when complete agreement with the working dataset is achieved, the $R_{\text{free}}^{\text{noise}}$ at the end of reconstruction instead decreases to a minimum and then starts to increase slightly (Fig. 2d). Such a trend was consistently observed with cross-validation sets selected from 10% to 30% of the overall measurements (Supplementary Fig. 6), highlighting the

robustness of the $R_{\text{work}}^{\text{noise}}$ and $R_{\text{free}}^{\text{noise}}$ measurements. Importantly, the change in $R_{\text{free}}^{\text{noise}}$ is closely mirrored by the change in the $\lambda$-dependent RMSD between the reconstruction and reference spectra (Fig. 2e): the minimal difference is achieved when $R_{\text{free}}^{\text{noise}}$ is at or very close to its minimum (Fig. 2f,g), but not when a very large value of $\lambda$ enforces complete agreement with the experimentally measured working dataset (Fig. 2h,i). These results strongly support the notion that $R_{\text{free}}^{\text{noise}}$ is a valid measure of the reconstruction spectral fidelity in the absence of an external reference spectrum.

We next tested the maximum entropy reconstruction method (MaxEnt)[10,11] using the same set of 1-D simulated data. The Fourier transforms of the sparsely sampled measurements and full measurements are shown in Fig. 3a,b, respectively. Reconstruction with the maximum entropy method is conceptually similar to $l_1$-norm minimization, except that the regularization term maximizes the information entropy $E(S)$ of the frequency domain spectrum rather than minimizing its $l_1$-norm. Mathematically, the optimal solution is achieved through a constrained minimization of the negative entropy $-E(S)$ and the Lagrangian-multiplier-weighted RMSD of the modelled time domain data $s$ and raw measurements $m$:

$$C = -E(S) + \lambda \times \text{RMSD}(s - m). \tag{3}$$

In the limit of the experimental noise being much larger than unity, a common condition encountered experimentally, the behaviour of the maximum entropy method is similar to the convex $l_1$-norm minimization algorithm. As the Lagrangian multiplier $\lambda$ was increased from zero, progressively more signals were included in the final reconstruction, resulting in an increase in the final composite score (Fig. 3c) and a decrease in the $R_{\text{work}}^{\text{noise}}$ and $R_{\text{free}}^{\text{noise}}$ (Fig. 3d). However, further increases in $\lambda$ caused more of the experimental noise to be included, resulting in a divergence of $R_{\text{work}}^{\text{noise}}$ and $R_{\text{free}}^{\text{noise}}$ (Fig. 3d). The change of the $R_{\text{free}}^{\text{noise}}$ over $\lambda$ was mirrored by the change of the RMSD of the reconstruction and reference spectra (Fig. 3e). A very large $\lambda$ ultimately enforced the agreement between modelled time domain data and the experimental measurements, a scenario that resembles the forward maximum entropy reconstruction[13] or maximum entropy interpolation[14]. It is important to note that the highest quality of reconstruction is achieved when $R_{\text{free}}^{\text{noise}}$ reaches a minimum (Fig. 3f,g), but not at a very large $\lambda$ (Fig. 3h,i).

As a final example, we applied these measures to the IST method, which enforces exact agreement between the reconstruction and the experimental data[12]. IST begins with the Fourier transform of the sparsely sampled working dataset. In each iteration, signals above a predefined threshold in the frequency domain are extracted (Fig. 4a), and their inverse Fourier transform is used to update the values of the time domain at all positions except those belonging to the working dataset. These extracted signals are, in effect, a model of the current reconstruction, and can be used to monitor $R_{\text{work}}^{\text{noise}}$ and $R_{\text{free}}^{\text{noise}}$. This process is repeated with a decreasing threshold, and the inverse Fourier transform of the model increasingly converges with the measured data of the working set. Snapshots of the IST reconstruction were taken at every step over 1,500 iterations. Although the model spectrum is typically not presented in iterations of spectral reconstruction by IST, it is intrinsic to the reconstruction process and is produced as an output in our implementation (Fig. 4a, right panel) and in the implementation of hmsIST[15], a variant of IST. Such a model spectrum allows meaningful calculation of both $R_{\text{work}}^{\text{noise}}$ and $R_{\text{free}}^{\text{noise}}$ (Fig. 4b).

As with other reconstruction methods, the $R_{\text{work}}^{\text{noise}}$ calculated from the model spectrum continued to decrease with an increasing number of IST iterations, eventually reaching zero; whereas $R_{\text{free}}^{\text{noise}}$ decreased to a minimum, then slightly increased

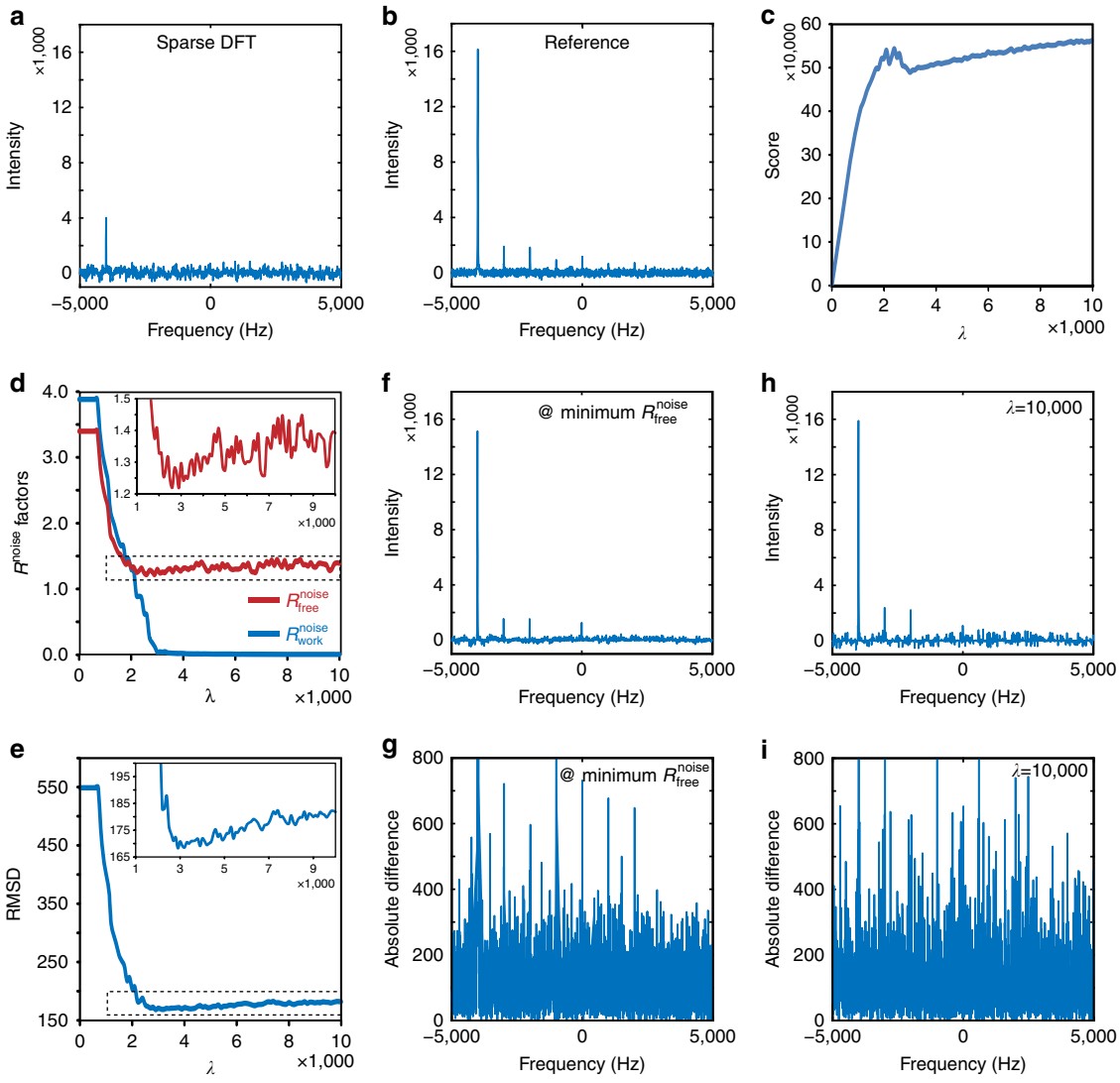

**Figure 2 | Correlation of $R_{free}^{noise}$ with the fidelity of spectral reconstruction by convex $l_1$-norm minimization.** (**a**) The Fourier transform of a sparsely sampled 1-D spectrum. (**b**) The reference spectrum containing the Fourier transform of a fully sampled 1-D spectrum. (**c**) Change in the composite score at the end of reconstruction for $l_1$-norm minimization as a function of the Lagrangian multiplier $\lambda$. (**d**) Change in the $R^{noise}$ factors at the end of reconstruction for $l_1$-norm minimization as a function of the Lagrangian multiplier $\lambda$. The inset shows a magnification of the boxed region. (**e**) The RMSD between the reconstruction and reference spectra as a function of the Lagrangian multiplier $\lambda$. The inset shows a magnification of the boxed region. (**f**) The reconstruction spectrum for the $\lambda$ that achieves the minimum $R_{free}^{noise}$. Its absolute difference spectrum from the reference spectrum is shown in **g**. (**h**) The reconstruction spectrum with a large Lagrangian multiplier ($\lambda = 10,000$). Its absolute difference spectrum from the reference spectrum is shown in **i**.

and eventually reached a plateau (Fig. 4b). The initial simultaneous decrease in $R_{work}^{noise}$ and $R_{free}^{noise}$ is consistent with the efficient processing of genuine signals, whereas the ultimate divergence of $R_{work}^{noise}$ and $R_{free}^{noise}$ reflects overfitting.

While the modification of IST to produce a model allows both measures of $R_{work}^{noise}$ and $R_{free}^{noise}$ to be used, we show that it is also possible to use $R_{free}^{noise}$ alone as a quality measure for the reconstructions generated by the unmodified IST algorithm. In the unmodified IST algorithm, $R_{work}^{noise}$ would not be a useful measurement of the spectral quality: when calculated from the reconstruction rather than the model, the value of $R_{work}^{noise}$ remains at zero during the entire run, reflecting the fact that the algorithm enforces exact agreement between the reconstruction and the measured data (Fig. 4c). However, as the reconstruction has no bias toward the free dataset, the $R_{free}^{noise}$ calculated from the evolving reconstruction spectrum behaves identically to that of the model spectrum (compare Fig. 4b,c), reinforcing the notion

that $R_{free}^{noise}$ is a universally applicable cross-validation measurement of reconstruction quality and model overfitting. Importantly, we again observe an excellent correlation between the RMSD curve of the reconstruction and reference spectra and the $R_{free}^{noise}$ curve (compare Fig. 4c,d): the highest quality of reconstruction is achieved when $R_{free}^{noise}$ is close to its minimum (Fig. 4e,f), and quality decreases with additional iterations (Fig. 4g,h). Further iterations beyond the minimum of $R_{free}^{noise}$ model more noise and artifacts than genuine signals (compare Fig. 4e,g)—a situation of model overfitting that leads to an increase of $R_{free}^{noise}$ (Fig. 4c) and the degradation of reconstruction fidelity (compare Fig. 4f,h).

## Discussion

Sparse sampling and iterative spectral reconstruction techniques are poised to transform magnetic resonance measurements in the

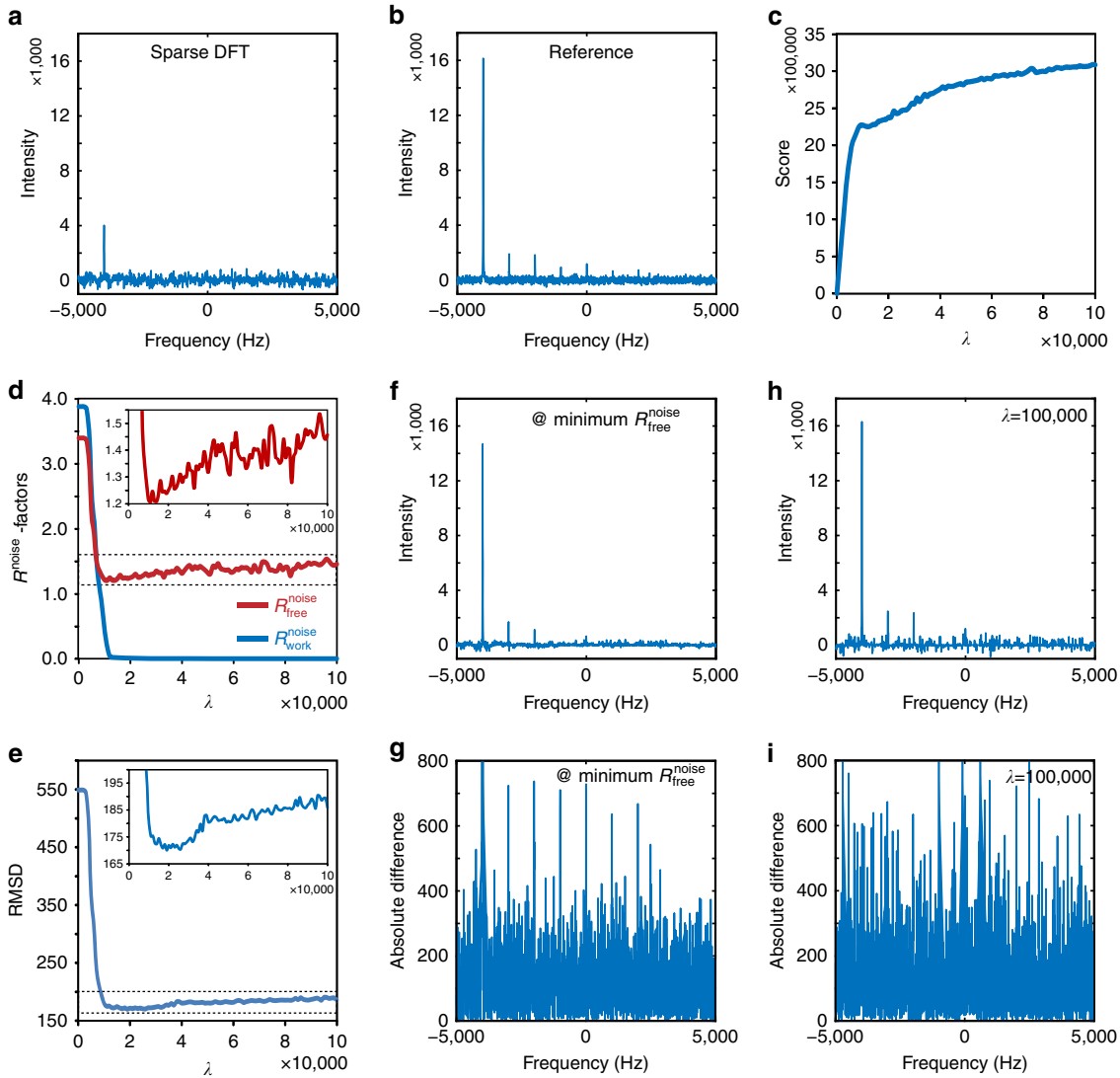

**Figure 3 | Correlation of $R_{free}^{noise}$ with the fidelity of spectral reconstruction by the maximum entropy method.** (**a**) The Fourier transform of a sparsely sampled 1-D spectrum. (**b**) The reference spectrum containing the Fourier transform of a fully sampled 1-D spectrum. (**c**) Change in the composite score at the end of reconstruction for the maximum entropy method as a function of the Lagrangian multiplier $\lambda$. (**d**) Change in the $R^{noise}$ factors at the end of reconstruction for the maximum entropy method as a function of the Lagrangian multiplier $\lambda$. The inset shows a magnification of the boxed region. (**e**) The RMSD between the reconstruction and reference spectra as a function of the Lagrangian multiplier $\lambda$. The inset shows a magnification of the boxed region. (**f**) The reconstruction spectrum for the $\lambda$ that achieves the minimum $R_{free}^{noise}$. Its absolute difference spectrum from the reference spectrum is shown in **g**. (**h**) The reconstruction spectrum with a large Lagrangian multiplier ($\lambda=100,000$). Its absolute difference spectrum from the reference spectrum is shown in **i**.

post-FT era[16], yet the characteristics and relative performance of the various reconstruction algorithms vary dramatically[17–19], demanding quantitative measurements for estimating reconstruction fidelity, for detecting inadequate modelling and for preventing model overfitting. It is clear from our 1-D simulations that algorithm-specific measures such as the composite score of convex $l_1$-norm minimization or maximum entropy are dependent on the reconstruction parameters (for example, the Lagrangian multiplier $\lambda$) and cannot be compared with each other directly for evaluation of reconstruction quality, whereas other algorithms, such as the IST, do not have a regularization score at all. The introduction of the cross-validation parameter $R_{free}^{noise}$ provides a benchmark for assessment of the reconstruction fidelity independent of reconstruction algorithm specifics. For algorithms that do not enforce exact agreement at measured time domain positions, $R_{work}^{noise}$ and $R_{free}^{noise}$ can be used jointly to identify inadequate

modelling and overfitting, while $R_{free}^{noise}$ is a universally applicable gauge of reconstruction fidelity in the absence of a reference spectrum.

The notion of cross validation has been used previously in compressed sensing for estimating decoding errors[20], and the method of using permutations of subsets of the raw data has also been used to search for convergent spectral reconstructions in NMR[21–23]. However, the cross-validation measures $R_{work}^{noise}$ and $R_{free}^{noise}$ presented in this work are novel, and are uniquely suited for application to NMR spectra.

The issue of model overfitting has been raised by Hoch and colleagues[14,18], though no algorithm-independent measurements for such effects have been reported. The development of $R_{free}^{noise}$ permits unbiased comparison of reconstruction methods and sampling patterns and a direct measurement of model overfitting. For example, a comparison of the three reconstruction algorithms in our 1-D simulation shows that the lowest $R_{free}^{noise}$ score was

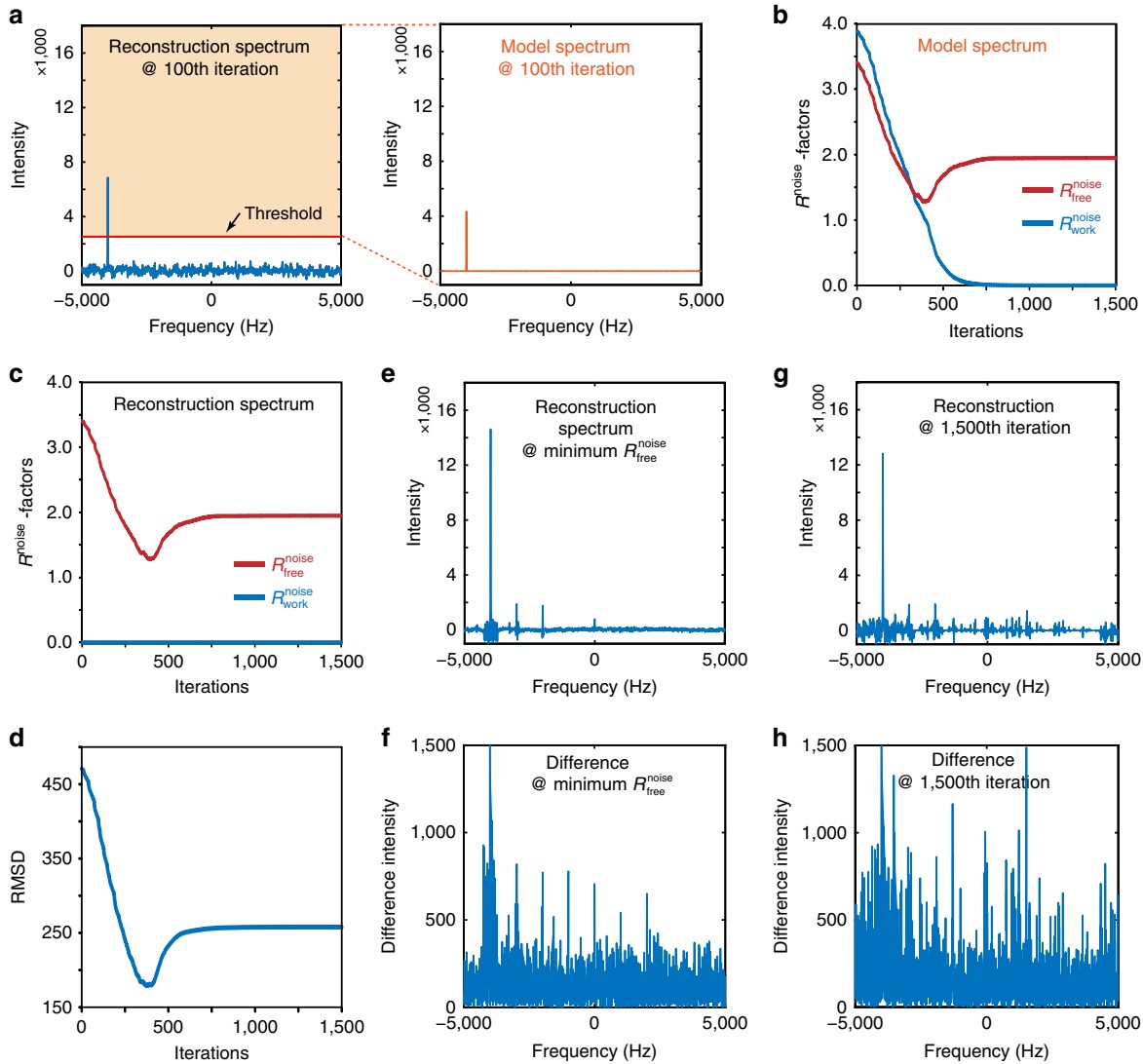

**Figure 4 | Correlation of $R_{free}^{noise}$ with the fidelity of spectral reconstruction by the IST method.** (**a**) Separation of the model spectrum from the reconstruction spectrum by IST. The left panel shows the IST reconstruction spectrum at the 100th iteration. The threshold for selecting modelled signals is indicated by a red line. Modelled signals (coloured area in the reconstruction spectrum) are replotted to generate the model spectrum (right panel). (**b**) Changes in $R_{work}^{noise}$ and $R_{free}^{noise}$ during iterations of IST for the model spectrum. (**c**) Changes in $R_{work}^{noise}$ and $R_{free}^{noise}$ during iterations of IST for the reconstruction spectrum. (**d**) Changes in the RMSD between the reconstruction spectrum and the reference spectrum during iterations of IST reconstruction. The reference spectrum is the same as in Figs 2b and 3b, and is not replotted here. (**e**) The IST reconstruction spectrum at the iteration with minimum $R_{free}^{noise}$. Its absolute difference spectrum from the reference spectrum is shown in **f**. (**g**) The IST reconstruction spectrum after 1,500 iterations. Its absolute difference spectrum from the reference spectrum is shown in **h**.

achieved by the convex $l_1$-norm minimization and maximum entropy methods, but only with the appropriate Lagrangian multipliers. Such a low $R_{free}^{noise}$ measurement is accompanied by the lowest RMSD between the reconstruction spectrum and the reference spectrum and thus the highest reconstruction fidelity. Reconstruction with less optimal Lagrangian multipliers leads to deterioration of the reconstruction fidelity either due to inadequate modelling or overfitting. The reconstruction fidelity of the IST method comes very close to the convex $l_1$-norm minimization and maximum entropy method when using an optimal number of iterations. However, the IST reconstruction is significantly worse with an infinite number of iterations, due to spectral overfitting.

As NMR spectral reconstruction is done independently for individual planes or cubes along the directly observed dimension, the most informative assessment of the reconstruction quality would be to calculate the quality factors separately for each position on the directly observed dimension. It is, however, conceivable that an overall quality factor could be calculated for the entire multi-dimensional spectrum, as the mean and standard deviation of the quality factors of the individual reconstructions.

The increasing sensitivity brought about by innovation in NMR instrumentation and pulse sequence design and the demand for more efficient data collection in biomolecular NMR studies have led to the burgeoning development of sparse sampling and reconstruction methods. The introduction and demonstration of the algorithm-independent reconstruction quality measurement $R_{free}^{noise}$ should provide much-needed quality assurance and greatly facilitate the wide adoption of sparse-sampling techniques in magnetic resonance spectroscopy.

## Methods

**NMR measurements, simulation and spectral reconstruction.** Three-dimensional sparsely sampled HNCO experiments were recorded on Agilent or Bruker

NMR spectrometers using $^{15}$N/$^{13}$C-labelled GB1, FAAP20 UBZ4, foldon, ubiquitin, the UBM1-ubiquitin complex and FKBP12. 2-D cosine-weighted randomized concentric ring sampling patterns[24] of 314 points adapted to the $64 \times 96$ sampling grid or 220 points adapted to the $64 \times 64$ sampling grid for indirect (N-C) dimensions were used, corresponding to a sampling density of $\sim 5\%$. A randomly selected dataset containing 90% of the measurements were used for spectral reconstruction via the CLEAN algorithm[8] and for calculation of $R_{work}^{noise}$ and $R_{work}$, whereas the remaining 10% measurements were excluded from reconstruction and were used for calculation of $R_{free}^{noise}$ and $R_{free}$. Modelled components of the CLEAN reconstruction were inverse Fourier transformed for comparison with the time domain measurements and for calculation of $R$-factors as described in the main text.

One-dimensional simulations were performed using MATLAB (MathWorks). The simulation contained nine exponentially decaying signals with amplitudes from 64 to 1 and frequencies from $-4,000$ to $4,000$ Hz in the presence of white noise. A pure noise dataset was also generated containing white noise of the same amplitude as the reference noise. A sampling grid of 1,024 points was used. A sparse dataset was created by randomly selecting 30% of the sampling points. Of this dataset, 80% of the measurements were used for spectral reconstruction and for calculation of $R_{work}^{noise}$, while the remaining measurements (20%) were excluded from spectral reconstruction and were only used for calculation of $R_{free}^{noise}$. Reconstruction was carried out using the convex $l_1$-norm minimization algorithm, the maximum entropy method, and iterative soft thresholding method. For assessing the stability of $R_{work}^{noise}$ and $R_{free}^{noise}$, additional tests were carried out with convex $l_1$-norm minimization using 90–70% of the measurements for spectral reconstruction and 10–30% for cross validation.

**Data availability.** NMR measurements and the software for calculating the $R^{noise}$ and $R$-factors and for 1-D simulations are available upon request from the authors.

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

## Acknowledgements

This work is supported by National Institutes of Health Grants AI094475 and GM115355 to P.Z. We would like to thank Dr Ronald Venters for sharing the FKBP12 data and for insightful comments on the manuscript.

## Author contributions

P.Z. conceived the study. Q.W. prepared samples and collected NMR data. All authors were involved in experimental design, data analysis and manuscript preparation.

## Additional information

**Competing financial interests:** The authors declare no competing financial interests.

