## [Peer Review File · Nature Communications]

Editorial Note: this manuscript has been previously reviewed at another journal that is not operating a transparent peer review scheme. This document only contains reviewer comments and rebuttal letters for versions considered at Nature Communications. Parts of this peer review file have been redacted to remove third-party material where no permission to publish was obtained.

Reviewer #1 (Remarks to the Author):

I very much appreciate work performed by the authors for the last revision of the manuscript. Finally, all the claims in the paper are properly explained and substantiated. I suggest publishing the manuscript as is.

Reviewer #2 (Remarks to the Author):

I hear the authors' desire to provide measurements of goodness regarding reconstructions of non-uniformly sampled (NUS) NMR spectroscopy. The issue of "overfitting" is however not a novel concern, but has been a quest of Jeffrey Hoch et. al. for some time. Hoch et al's approach ought to be acknowledged within the manuscript and not just via references! (see e.g : Paramasivam et al, J Phys Chem B. 2012 Jun 28; 116(25): 7416-7427.)

The deeper question is that of "correctness" and usefulness. To this I would like to allude to the situation of the so-called "Milkmaid problem" often associated to the introduction of Lagrange Multipliers. E.g. <http://www2.sjs.org/raulston/mvc.10/lagrange.intro.htm>. The addition of λ lets us solve the problem in an efficient way; yet at the end of the solution, λ is always eliminated.

However, if we somewhat change the Milkmaid problem with adding that the water level of the river tend to change, and hence flood the meadow in various unpredictable ways. Then it is a question which of following solutions is "correct", (i) assuming the maximum height of the river, (ii) the average height of the river, or (iii) the average height of the river plus one standard deviation. The solutions (i) and (iii) retains the Lagrange λ with a finite value.

The practical aspect with all reconstruction of NUS is that the problem is underdetermined and a model for solution has to be entered. The problem hence becomes non-linear. This is manifested in NUS reconstructed spectra that (a) the reconstruction of the noise becomes non-Gaussian (non-white), and (b) an absolute attenuation of the signal intensities that depends on the intrinsic noise and the sparsity. Notable is that the precession of the signal height is still Gaussian distributed.

Due to the attenuation, the relative signal height is perturbed in favor of strong signals. This effect is documented (Hyberts et al, J Biomol NMR (2013) 55:167-178). These effects are even more pronounced when compensating for "overfitting" (i.e. retaining the the Lagrange λ with a finite value). The spectra where compensation is done are hence possibly less useful as they in essentially add the attenuation with the width of precision, reducing the signal heights further and effectively aggravating the problem with relative signal heights further.

Please also note that in the above-mentioned publication, a procedure of determining estimating sensitivity of NMR spectra. This procedure is then also applied to NUS obtained simulations. As this is done with hmsIST to conversion (i.e., considered to be "overfitted"), artifacts would be degrading the results if they would be serious. The procedure evaluates if the largest point in the spectrum is at the position of the known simulated signal. If elsewhere (i.e., the procedure picks up an artifact), the detection probability is lessened. It is however found that the detection probability is mainly the same for the non-weighted sampling (SSW=0 in figure 4A) as it is for the uniformly sampled case. Hence it is unlikely that the issue of artifacts is of any severity. This is that the problem of "overfitting" NUS reconstruction may be considered minor.

Remark 1 to previous Reviewer 1 and response:

It is true that one can state that FM is a special case of IST (or rather the other way as FM is able to use other target functions than ℓ_1 -norm), however only when both have converged. During the process, there is however a distinction, which Reviewer #1 eludes to, namely that, the obtained data points are kept static throughout the minimization of FM, and not throughout the process of IST. The curves of that are produced over the iterations of IST would hence most likely be very different than those created by FM with different numbers of iterations.

Remark 2 to previous Reviewer 2 and response:

I am not convinced that the conclusion of $R_{\text{noise/work}}$ is as much a universal measurement of the reconstruction quality as it is a measurement of the spectral situation. Two additional situations would be appreciated to be investigated are:

1) As IST can very well be applied to traditional uniformly sampled spectra, it would be very interesting to see if there is a "quality" cutoff in similar manner with finite iterations. It would evaluate the universality of the definition of these parameters.

2) At page 5 in the manuscript it is stated "In NMR, however, each position on the directly observed dimension of a spectrum constitutes an independent reconstruction problem...". Indeed, this is the common reconstruction procedure as it allows for a parallel approach. I.e. it allows for shorter reconstruction time by farming out the process on several independent computations. However as this manuscript claim universality and a measure of quality, there is no problem in considering the observed dimension as to be part of the reconstruction. This as I see it will alter the calibration of the noise.

Reviewer #2 (Remarks to the Author):

I hear the authors' desire to provide measurements of goodness regarding reconstructions of non-uniformly sampled (NUS) NMR spectroscopy. The issue of "overfitting" is however not a novel concern, but has been a quest of Jeffrey Hoch et. al. for some time. Hoch et al's approach ought to be acknowledged within the manuscript and not just via references! (see e.g. : Paramasivam et al, J Phys Chem B. 2012 Jun 28; 116(25): 7416-7427.)

In response to the reviewer's comment, the following sentence is added to the revised manuscript. "The issue of model overfitting has been raised by Hoch and colleagues^{14, 18}, though no algorithm-independent measurements for such effects have been reported."

The deeper question is that of "correctness" and usefulness. To this I would like to allude to the situation of the so-called "Milkmaid problem" often associated to the introduction of Lagrange Multipliers. E.g. <http://www2.sjs.org/raulston/mvc.10/lagrange.intro.htm>. The addition of λ lets us solve the problem in an efficient way; yet at the end of the solution, λ is always eliminated.

However, if we somewhat change the Milkmaid problem with adding that the water level of the river tend to change, and hence flood the meadow in various unpredictable ways. Then it is a question which of following solutions is "correct", (i) assuming the maximum height of the river, (ii) the average height of the river, or (iii) the average height of the river plus one standard deviation. The solutions (i) and (iii) retains the Lagrange λ with a finite value.

The practical aspect with all reconstruction of NUS is that the problem is underdetermined and a model for solution has to be entered. The problem hence becomes non-linear. This is manifested in NUS reconstructed spectra that (a) the reconstruction of the noise becomes non-Gaussian (non-white), and (b) an absolute attenuation of the signal intensities that depends on the intrinsic noise and the sparsity. Notable is that the precession of the signal height is still Gaussian distributed.

Due to the attenuation, the relative signal height is perturbed in favor of strong signals. This effect is documented (Hyberts et al, J Biomol NMR (2013) 55:167-178). These effects are even more pronounced when compensating for "overfitting" (i.e. retaining the the Lagrange λ with a finite value). The spectra where compensation is done are hence possibly less useful as they in essentially add the attenuation with the width of precision, reducing the signal heights further and effectively aggravating the problem with relative signal heights further.

Please also note that in the above-mentioned publication, a procedure of determining estimating sensitivity of NMR spectra. This procedure is then also applied to NUS obtained simulations. As this is done with hmsIST to conversion (i.e., considered to be "overfitted"), artifacts would be degrading the results if they would be serious. The procedure evaluates if the largest point in the spectrum is at the position of the known simulated signal. If elsewhere (i.e., the procedure picks

up an artifact), the detection probability is lessened. It is however found that the detection probability is mainly the same for the non-weighted sampling ($SSW=0$ in figure 4A) as it is for the uniformly sampled case. Hence it is unlikely that the issue of artifacts is of any severity. This is that the problem of "overfitting" NUS reconstruction may be considered minor.

We thank the reviewer for the insightful comments. The pros and cons of existing reconstruction methods are beyond the scope of this study. It is sufficient to say that a great number of algorithms have been developed for reconstructing sparsely sampled NMR spectra with vastly different outcomes. To illustrate this point, we present below two examples that are adapted from **Figure 7** in *Journal of Biomolecular NMR* 2010; 47: 65-77 and from **Figure S1** in *Biochemical and Biophysical Research Communications* 2015; 457: 200-205 (**Figure 1**). Both are reconstructions of 3-D NOESY spectra using *the same input data*, but with different reconstruction algorithms. The reference spectra were purposely removed from the original figures so that the reviewer can evaluate the reconstructed spectra without bias.

Editorial Note: Parts of this peer review file have been redacted to remove third-party material where no permission to publish was obtained.

Figure 1. Different qualities of the reconstructed 3-D NOESY-HSQC spectra from the same input data. (A) Adapted from **Figure 7b-c**, *Journal of Biomolecular NMR* 2010; 47: 65-77. (B) Adapted from the middle panel of **Figure S1** from *Biochemical and Biophysical Research Communications*, 2015; 457: 200-205. The reference spectra of panels (A) and (B) are purposely removed from the original publications to avoid bias for signal identification. Boxed areas in panel (A) include reconstruction artifacts that are indistinguishable from the real signals and illustrate the consequence of reconstruction artifacts on data interpretation.

There are two points we would like to emphasize here. First, with the *same input data*, different reconstruction algorithms have yielded strikingly different reconstruction spectra. Which one is the best representation of the real spectrum? Is it the one with the largest number of signals, the one with the smallest number of signals, or the one in between? How do we reach such a conclusion? On what basis do we reach such a conclusion?

Second, we would like to ask the reviewer to focus on the boxed areas in the left panel. Most of these signals are missing in the maximum entropy reconstruction (ME), but are present in the reconstructions by MDD and SSA/FT. Yet, the latter two algorithms do not yield consistent signals either. Some of the reconstructed signals in the boxed regions are artifacts (based on comparison with a fully sampled reference spectrum). Can the reviewer tell which ones are genuine signals and which ones are artifacts in the absence of a fully sampled reference spectrum? We cannot.

The above examples illustrate that (1) the quality of reconstructed NMR spectra can vary greatly even *with the same input data*; (2) reconstruction artifacts can cause serious issues for NMR data interpretation; and (3) it is absolutely necessary to establish a fidelity measurement independent of the reconstruction algorithm for detection of inadequate modeling as well as overfitting, which is the focus of this study.

1) As IST can very well be applied to traditional uniformly sampled spectra, it would be very interesting to see if there is a "quality" cutoff in similar manner with finite iterations. It would evaluate the universality of the definition of these parameters.

The calculation of the R_{free}^{noise} factor requires randomly selecting a subset of the data for cross-validation. Once this is done, the remaining data would become non-uniformly sampled, and there is no difference from the examples already illustrated in the manuscript.

2) At page 5 in the manuscript it is stated "In NMR, however, each position on the directly observed dimension of a spectrum constitutes an independent reconstruction problem...". Indeed, this is the common reconstruction procedure as it allows for a parallel approach. I.e. it allows for shorter reconstruction time by farming out the process on several independent computations. However as this manuscript claim universality and a measure of quality, there is no problem in considering the observed dimension as to be part of the reconstruction. This as I see it will alter the calibration of the noise.

While we agree with the reviewer that it is possible to include the direct dimension in the reconstruction, such an approach is either impractical (for 4-D data) or undesirable (for 3-D data) due to the tremendous cost of memory and speed. There is also little benefit for the suggested approach as reconstructions of individual positions of the direct dimension are completely independent events. For these reasons, none of the reconstruction algorithms takes the approach

suggested by the reviewer. Instead, all of the reconstructions are done for individual planes or cubes along the directly observed dimension. As such, our quality measurement should certainly reflect this – i.e., the quality factors should be calculated for individual reconstructed planes and cubes as presented in this manuscript.

It is however conceivable that an overall quality factor could be calculated for the entire 3-D or 4-D spectrum, as the mean and standard deviation of individual planes or cubes along the directly observed dimension. This is commented on in the revised manuscript. Such a practice, however, does not change the calculation of the noise level.